# Once-for-all: Efficient Visual Face Privacy Protection via Person-specific Veils

## ABSTRACT

As billions of face images stored on cloud platforms contain sensitive information to human vision, the public confronts substantial threats to visual face privacy. In response, the community has proposed some perturbation-based schemes to mitigate visual privacy leakage. However, these schemes need to generate a new protective perturbation for each image, failing to satisfy the real-time requirement of cloud platforms. To address this issue, we present an efficient visual face privacy protection scheme by utilizing person-specific veils, which can be conveniently applied to all images of the same user without regeneration. The protected images exhibit significant visual differences from the originals but remain identifiable to face recognition models. Furthermore, the protected images can be recovered to originals under certain circumstances. In the process of generating the veils, we propose a feature alignment loss to promote consistency between the recognition outputs of protected and original images with approximate construction of feature subspace. Meanwhile, the block variance loss is designed to enhance the concealment of visual identity information. Extensive experimental results demonstrate that our scheme can significantly eliminate the visual appearance of original images and almost has no impact on face recognition models.

## CCS CONCEPTS

• **Security and privacy → Privacy protections**; • **Computing methodologies → Image representations**;

## KEYWORDS

Cloud platforms, Visual face privacy, Adversarial perturbation, Face recognition, Feature subspace

## 1 INTRODUCTION

The emergence and development of deep neural networks (DNNs) provide powerful tools for addressing complex computer vision tasks, e.g., face recognition (FR) [8, 23], pose estimation [46], and medical image analysis [13]. In the past years, DNN-based FR applications have been adopted in various domains. For example, Google Photos and Amazon Rekognition enable users to effortlessly identify and search for specific individuals within a given collection of photos in real-time. In the law enforcement system, FR is utilized to assist in criminal investigations [40].

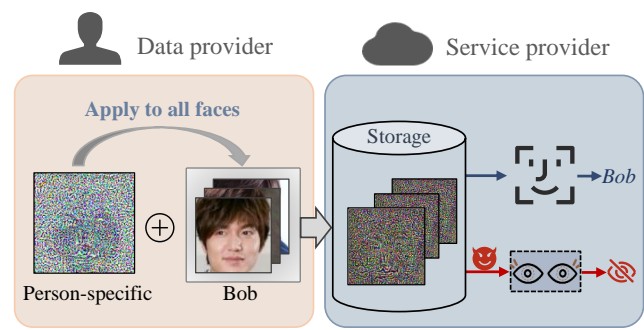

**Figure 1: A paradigm of our scheme applied in cloud-based FR services. A data provider applies a person-specific veil to all face images of Bob to protect them. These protected images are identifiable to an FR model while being concealed to human vision. Due to the unnecessity of crafting new perturbations multiple times, our scheme is convenient for a user and satisfies the real-time capability of an FR service.**

Considering a holistic perspective on cost-effectiveness and implementation convenience, these applications are usually offered by third-party cloud service providers. It implies that the unprocessed face images containing sensitive visual information, *e.g.,* facial features, health status, and cultural background, are directly exposed to the naked eyes of these cloud service providers or malicious attackers. This visual information could potentially be exploited for false identity authentication or unlawful surveillance, which poses a visual face privacy threat to the public [5, 39, 44]. Therefore, it is imperative and urgent to explore effective schemes to minimize the leakage of sensitive visual information within face images.

Proposals in some studies [3, 4, 22, 42] are made to leverage homomorphic encryption (HE) techniques to mitigate the exposure of visual content. Meanwhile, FR can be directly performed in the encrypted domain without retraining a new model. Nonetheless, due to the non-linear activation function employed by the advanced DNNs, these HE-based schemes incur significant performance degradation, which is scarcely applicable to real-time FR services. Furthermore, although several differential privacy-based schemes [7, 12, 24] can achieve a satisfactory FR accuracy rate and guarantee visual privacy, the inherent drawback of differential privacy algorithms limits their user-friendliness and scalability.

Recently, a few researchers have shown a keen interest in utilizing adversarial attacks to protect visual face privacy. Generally, adversarial attacks refer to the employment of quasi-imperceptible and elaborate tactics, named adversarial perturbation, to mislead target models [36]. Different from the traditional adversarial perturbations mentioned above, some works [28, 33] attempt to explore the positive impacts of adversarial perturbations in visual face privacy protection. The AVIH [33] generates the protected image by

transforming the benign image to random noise while maintaining the high similarity between the identity feature vectors of both. The visually-protected images retain the functionality of FR and can be recovered as the original images with a correct key. Liu *et al.* [28] proposed a RIC system, which can encode plaintext images into noise-like adversarial examples (NAEs). Such NAEs can be correctly classified by plaintext-domain classifiers with a high accuracy rate, and support high-fidelity recovery. However, existing perturbation-based schemes need to separately craft a protective perturbation for each image (image-specific) and ignore the intimate relationships between individual images, namely the similarity in appearance and feature vectors among these images. Consequently, they may meet a significant expenditure of computational and storage resources, particularly when handling extensive collections of face images, thereby constraining the practical applicability in real-world scenarios.

To break the limitations mentioned above, we propose an efficient visual face privacy protection scheme by using person-specific veils. In our scheme, a user can conveniently apply the veil to all his/her images, which removes the need to generate new perturbations multiple times. These protected images eliminate visual identity information from the naked eye of any human observer (visual anonymization) but can be directly recognized as the original output by the existing FR model (identity preservation). Furthermore, the number of veils is solely determined by the number of users rather than images, which means that our scheme greatly reduces the consumption of computational resources for visual face privacy protection. Compared with previous schemes, our scheme takes an important step toward a practical system with real-time requirements. There is an application scenario of our scheme in the cloud environment in Fig. 1.

Specifically, for the generation of person-specific veils, we first construct an identity feature subspace based on the similarity of appearance and feature vectors among images of the same identity. Secondly, we minimize the distance between the feature vectors of the protected images and their corresponding feature subspace. Meanwhile, we also minimize the Euclidean distance between the original and protected images to increase their visual perceptual differences. Thirdly, to further enhance the concealment of visual identity information, we design a block variance loss to reduce the differences between adjacent pixels in the protected images. It is worth noting that the protected images are formed by merging benign images with a person-specific veil at a certain proportion. Therefore, when authorized users obtain the veil under certain circumstances, they can recover the benign images from the protected ones.

Our main contributions are summarized as follows:

- We develop an efficient visual face privacy protection scheme by utilizing person-specific veils. In our scheme, a user can apply such a veil to all his/her images to hide visual identity, without crafting new veils multiple times.
- We design an optimization-based method to craft person-specific veils. Our method fosters consistency between the recognition outputs of protected and original images by employing the feature subspace and enhances the concealment of visual identity information with block variance loss.

- Extensive qualitative and quantitative experiments to demonstrate the effectiveness of our scheme on visual anonymization and identity preservation. Furthermore, we also perform an ablation study to present the effectiveness of the block variance loss.

## 2 RELATED WORK

### 2.1 Adversarial Attack

Since Szegedy *et al.* [36] found the vulnerability of DNN to adversarial attack, there has sparked an increasing interest and research focus on this topic [1, 2]. Generally, adversarial attacks involve utilizing quasi-imperceptible and elaborate tactics, known as adversarial perturbation, to deceive target models. Such sort of adversarial attack was termed adversarial attack Type II by Tang *et al.* [37] and they also defined adversarial attack Type I. In brief, adversarial attack Type I aims to seek an input image significantly different from the benign image while ensuring that the outputs of the target models remain the same as possible.

There is a large amount of research on adversarial attack Type II, among which, an influential emphasis is placed on the exploration of gradient-based methods. As a seminal work, the fast gradient sign method (FGSM) [15] has emerged as a prominently investigated technique. FGSM utilizes the gradient information calculated by the target model's loss function against the input image to craft adversarial perturbation. Subsequently, there has been a surge of follow-up works [10, 11, 26, 41]. Differing from the above-mentioned methods that are solely applicable to a single image, Moosavi-Dezfooli *et al.* [30] discovered the existence of the universal adversarial perturbation (UAP). Building on this, Gupta *et al.* [17] introduced class-wise UAPs, and Zhang *et al.* [43] proposed CD-UAPs, which is a variant of the former. Moreover, Zhong *et al.* [45] applied class-wise UAPs to the field of face privacy protection and developed OPOM.

Compared to adversarial attack Type II, the research on adversarial attack Type I is at an initial stage [18, 34, 37]. Tang *et al.* [37] introduced the basic definition of adversarial attack Type I and used it to conduct preliminary experiments against FR and image classification. Subsequently, Sun *et al.* [34] employed it against generative models such as VAEs. He *et al.* [18] further utilized the distribution mapping from the source domain to the target domain to control the directional visual transformations of the protected images.

### 2.2 Visual Face Privacy Protection

Visual face privacy protection means that the protected images are unrecognizable to human vision. Although previous methods, e.g., blurring [14], mosaicing [6], and occlusion [29], can protect visual face privacy to a certain extent possibly, they compromise the identity recognition utility of the protected images. In other words, the protected images fail to be directly employed in real-time FR services, which limits the applicability of the schemes [32]. Therefore, we will not discuss these types of visual face privacy protection schemes mentioned above in this subsection.

For the purpose of safeguarding visual face privacy and maintaining the identity recognition utility of the protected images, a feasible solution is homomorphic encryption (HE) [3, 4, 22, 42]. These HE-based schemes enable FR to be performed directly on

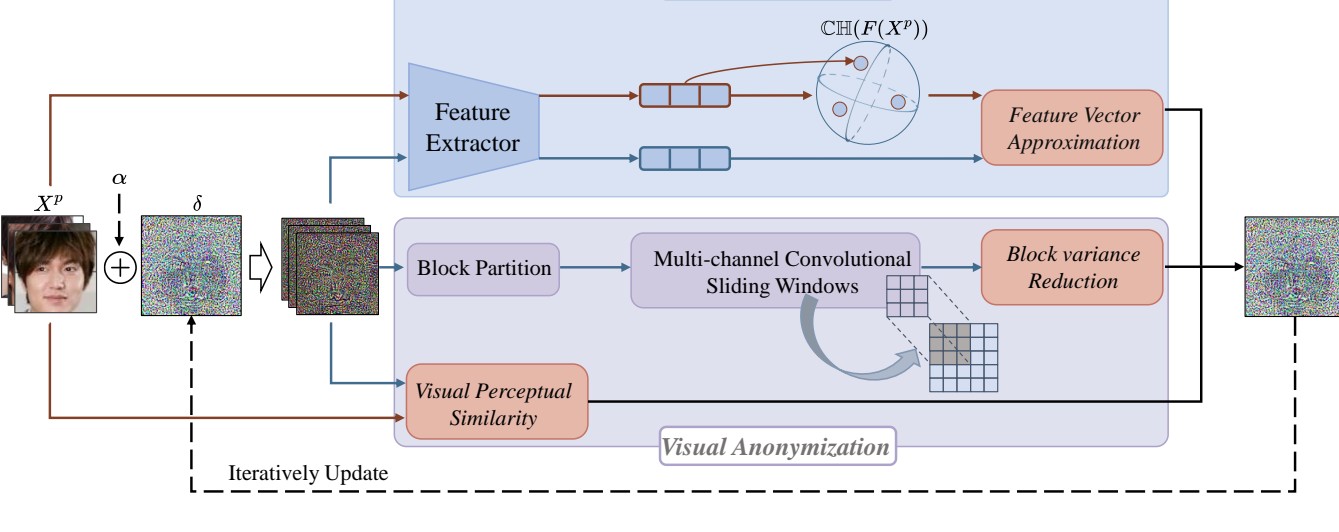

**Figure 2: The pipeline of our proposed method. The protected images are generated with a set of face images $X^p$ of the same identity and a person-specific veil $\delta$. $\mathbb{CH}(F(X^p))$ denotes the identity feature subspace of the original images.**

the encrypted images without decryption. However, when these schemes are applied to the advanced DNNs, the performance of FR will degrade significantly. To address this problem, some differential privacy-based schemes were proposed [7, 12, 24]. A notable one is DCTDP [24], which protects visual face privacy with learnable privacy budgets based on frequency domain differential privacy. Moreover, Ji *et al.* [24] showcased the satisfactory recognition accuracy of the protected images against the target FR model. Unfortunately, it is the inherent drawback of differential privacy algorithms that limits their user-friendliness and scalability.

Recently, a few researchers have shown a keen interest in utilizing adversarial attack Type I to protect visual face privacy. Su *et al.* [33] introduced AVIH, which generates the protected image by transforming the benign image to random noise while maintaining the high similarity between the identity feature vectors of both. The visually protected images retain the functionality of FR and can be recovered as the original ones. In a similar vein, Liu *et al.* proposed an RIC system, which can encode plaintext images into Noise-like Adversarial Examples (NAEs). These NAEs can be correctly classified by plaintext-domain classifiers with a satisfactory success rate, and support high-fidelity recovery. Nonetheless, existing perturbation-based schemes need to separately craft a protective perturbation for each image. Therefore, when the number of images and users increases, exponential computation and storage resources will be consumed, which is not user-friendly and impractical. In this paper, we expect to employ the foundational concepts of adversarial attack Type I and class-wise UAPs to protect visual face privacy.

## 3 THE PROPOSED SCHEME

### 3.1 Problem Formulation

Let $F(x) : \mathbb{X} \to \mathbb{R}^d$ denotes a FR model $F$ that extracts a normalized feature vector in $\mathbb{R}^d$ for an input image $x \in \mathbb{X}$, and the output of $F$ is

symbolized $\hat{F}(x)$. Similarly, $H$ denotes the human vision system and the output of $H$ is represented as $\hat{H}(X)$. Given a limited number $n_p$ of face images $X^p = \left\{ x_1^p, x_2^p, x_3^p, ..., x_{n_p}^p \right\}$ of identity $p$, our scheme aims to craft a person-specific veil $\delta \in \mathbb{R}^d$, which provides an efficient visual face privacy protection. Once generated, the veil can be conveniently overlaid to all images in $X^p$. The protected images retain the functionality of FR and conceal the visual identity information to human vision. Therefore, according to adversarial attack Type-I and UAP mentioned above, the protected images ought to satisfy the following two properties.

**Property 1 (Identity Preservation).** *The protected images can be directly recognized as the original identity by the existing FR model without retraining:*

$$\hat{F}(x_i^p) = \hat{F}(x_i^p + \delta), for\ most\ x_i^p \in X^p. \tag{1}$$

The probability of equality between $\hat{F}(x_i^p)$ and $\hat{F}(x_i^p + \delta)$ reflects the extent to which the functionality of FR is preserved.

**Property 2 (Visual Anonymization).** *The visual identity information in the protected images is invisible to the naked eye of any human observer:*

$$\hat{H}(x_i^p) \neq \hat{H}(x_i^p + \delta), for\ most\ x_i^p \in X^p. \tag{2}$$

The distance between $\hat{H}(x_i^p)$ and $\hat{H}(x_i^p + \delta)$ reflects the effectiveness of visual identity concealment. In this paper, we aim for both aspects to achieve satisfactory performance.

### 3.2 The Pipeline of Our Method

The pipeline of our method is illustrated in Fig. 2. To generate the protected images, a user can craft a person-specific veil $\delta$ and apply it to all images in $X^p$. Such a veil is crafted by achieving two parallel objectives: identity preservation and visual anonymization. Regarding identity preservation, we attempt to minimize the distance between the features of the protected images and the original

identity feature subspace $\mathbb{CH}(F(X^p))$. As for visual anonymization, it is expected to increase the visual perceptual differences between the original and protected images. In addition, to enhance the concealment of visual identity information further, the block variance of the protected images should be reduced. The following will elaborate on the details of our method.

## 3.3 Identity Preservation

*3.3.1 The Construction of Identity Feature Subspace.* Due to a large set of images $X^p$ being often inaccessible in real scenarios, it is practical to craft the person-specific veils with a limited number of face images. Nonetheless, the quantity of provided images exerts a notable influence on the performance of the two aspects mentioned above.

To deal with this problem, we endeavor to analyze the close connections in all images belonging to the same identity. From the perspective of human vision, the images in $X^p$ exhibit similar visual profiles. From the standpoint of machine vision, these images yield the same recognition output. These two intuitive observations capture our interest, which prompts us to construct an approximate identity feature subspace for generating the person-specific veil. By minimizing the distance between the visually protected images $x_i^p + \delta$ and the corresponding identity feature subspace, we can reformulate Equation (1) as

$$\underset{\delta}{\arg\min} \, (D(F(X^p), F(x_i^p + \delta)), for \, most \, x_i^p \in X^p, \quad (3)$$

where $F(X^p)$ represents the feature subspace of identity $p$, and $D(\cdot)$ computes the distance between a feature representation and a feature subspace. The solution to this distance can be regarded as a best approximation problem. Recall that given a matrix $W \in \mathbb{R}^{m \times n}$ and a vector $\vec{y} \in \mathbb{R}^m$, the minimum distance of them can be described as

$$\underset{x^*}{\min} \|W\vec{x} - \vec{y}\|_2, \quad (4)$$

and the solution is

$$\vec{x}^* = (W^T W)^{-1} W^T \vec{y}, s.t. \, rank(W) = n, \quad (5)$$

where $rank(\cdot)$ represents the rank of the provided matrix and the matrix $W$ is invertible if and only if $rank(W) = n$.

*3.3.2 Feature Vector Approximation.* Note that the identity feature subspace is high-dimensional and irregular, we cannot describe it clearly and completely with mathematical formulas. Hence, we endeavor to seek an approach to approximately modeling this feature subspace with a subset of face images. Specifically, we define the approximation of the feature subspace of $p$ as

$$\mathbb{CH}(F(X^p)) = \left\{ \sum_{x_i^p \in X^p} w_i^p F(x_i^p) \right\}$$
$$s.t. \, w_i^p \geq 0, \sum_{x_i^p \in X^p} w_i^p = 1, \quad (6)$$

where $w_i^p (i = 1, 2, 3, ..., n_p)$ are the weight coefficients corresponding to features representations and show the contribution of each of those in the process of constructing $\mathbb{CH}(F(X^p))$. It is reasonable for $w_i^p \geq 0$ since the contribution of each feature representation

should not be less than zero. With Equation (10), we can define the feature alignment loss $L_{fea}$ as

$$L_{fea}(X^p) = D(\mathbb{CH}(F(X^p)), F(x_i^p + \delta), for \, most \, x_i^p \in X^p. \quad (7)$$

With Singular Value Decomposition, $\mathbb{CH}(F(X^p))$ can be factorized into a unitary matrix $U$, a diagonal matrix $\Sigma$, and another unitary matrix $W$. Furthermore, $U$ is the orthogonal basis of $\mathbb{CH}(F(X^p))$ and remains constant, which is spanned by $F(x_i^p)$. Therefore, the solution of the optimization problem in Equation (3) can be converted into finding appropriate weight coefficients. That is, in this way, we can employ Equation (5) to obtain the best approximation solution in $W$ for Equation (7).

## 3.4 Visual Anonymization

*3.4.1 Visual Perceptual Similarity.* As all we know, the human vision system is recognized to be highly complex and elusive. In this paper, consistent with most adversarial perturbation schemes, we conventionally employ the $l_2$ distance to simulate human visual perception. Therefore, we can define the visual perceptual loss $L_{vis}$ as

$$L_{vis}((x_i^p, x_i^p + \delta) = \left\| x_i^p - (x_i^p + \delta) \right\|, for \, most \, x_i^p \in X^p. \quad (8)$$

Since the visual identity information within the original images needs to be concealed, the $L_{vis}$ should be minimized. However, The simulation effectiveness of the $l_2$ distance has certain limitations, which will significantly impact the performance of visual anonymization.

*3.4.2 Block Variance Reduction.* As mentioned above, although the protected images obtained by Equation (8) exhibit significant visual perceptual differences from the originals, there still exist a few remnants of the face profile, which undermines the effectiveness of visual face privacy protection. To deal with this problem, we aim to minimize the differences in adjacent pixel values within the protected images. Specifically, for each JPEG image $x$, we first divide it in each channel (R, G, B) into b blocks with each block size being $h \times w$, i.e., $\{x_1, x_2, x_3, ..., x_b\}_c$, where $c \in \{R, G, B\}$. Then, we utilize a convolution kernel of the size of $h \times w$ ($3 \times 3$), with each element being 1, to perform sliding-channel convolution and obtain the sum of each block in each channel, i.e., $S_c = \{S_1, S_2, S_3, ..., S_b\}_c$. Next, we can calculate the block inner-variance $\text{Var}_{inner}(x)$ as

$$\text{Var}_{inner}(x) = \sum_{c \in \{R,G,B\}} \text{var}(S_c), \quad (9)$$

where **var** calculates the variance value of a given block and the block intra-variance $\text{Var}_{intra}(x)$

$$\text{Var}_{intra}(x) = \sum_{c \in \{R,G,B\}} \frac{1}{b} \sum_{i=1}^{b} \left( \mu_i - \frac{1}{b} \sum_{i=1}^{b} S_{ic} \right)^2, \quad (10)$$

where $\mu_i = \frac{S_{ic}}{h \times w}$. Finally, the total block variance loss $L_{var}$ in the protected images can be computed by

$$L_{var}(x_i^p + \delta) = \text{Var}_{inner}(x_i^p + \delta) + \text{Var}_{intra}(x_i^p + \delta). \quad (11)$$

Reviewing our goal, we expect to increase the difference between protected images and the originals. Therefore, the $L_{var}(x_i^p + \delta)$ should be minimized.

---

**Algorithm 1:** The Generation of Person-specific Veils

**Input:** A FR model $F$, face images $X^p = \{x_1^p, x_2^p, x_3^p, ..., x_{n_p}^p\}$ of identity $p$, maximum number of iterations $t_{\max}$, decay factor $\mu$.

**Output:** Person-specific veil $\delta_{t_{\max}}$ for identity $p$

1   Initialization $\delta_0 = x_1^p, t = 0, \mu = 1, g_0 = 0$;

2   **for** $t = 0$ *to* $t_{max} - 1$ **do**

3      Obtain $L_{total}$ via Equation (13);

4      Calculate the gradient $\nabla L_{total}$;

5      $g_{t+1} = \mu \cdot g_t + \frac{\nabla L_{total}}{\|\nabla L_{total}\|_1}$;

6      $x_i^{p\prime} = Clip(x_i^{p\prime}, 0, 255)$;

7      $\delta_i = \frac{1}{(1-\alpha)} \cdot (x_i^{p\prime} - \alpha \cdot x_i^p)$;

8   **end**

---

## 3.5 The Generation of Person-specific Veils

In addition, we expect to enable our scheme support for recoverability, which offers flexibility in managing privacy settings. Therefore, we modify the generation of the protected images as

$$x_i^{p\prime} = \alpha \cdot x_i^p + (1-\alpha) \cdot \delta, s.t. \ x_i^p \in X^p, \tag{12}$$

where $\alpha$ controls the proportion of the original image in the protected image. When $\alpha$ is small, the impact of the veil predominates, resulting in a larger visual perceptual difference between the protected image and the original image. Now, we can craft the veil $\delta$ by

$$\delta = \underset{\delta}{\arg\min} \ L_{total}$$
$$= \underset{\delta}{\arg\min} \sum_{x_i^p \in X^p} (L_{fea}(X^p) - \lambda \cdot L_{vis}(x_i^p, x_i^{p\prime}) + \mu \cdot L_{var}(x_i^{p\prime})), \tag{13}$$

where $\lambda$ and $\mu$ are hyperparameters to balance the performance of visual transformation and identity preservation.

**Recoverability.** In practice, each user can employ a unique initial value assigned by us or a trustworthy third-party institution to craft their own veil and can recover the original image by

$$x_i^{p\prime\prime} = \frac{1}{1-\alpha} \cdot (x_i^{p\prime} - (1-\alpha) \cdot \delta), \tag{14}$$

where $x_i^{p\prime\prime}$ is the recovered image. Furthermore, we employ Gaussian filtering to the recovered images to improve the visual quality. Finally, we incorporate MI-FGSM [10] into our method to improve the performance of visual face privacy protection. Our algorithm for generating person-specific veils is presented in Algorithm 1.

## 4 EXPERIMENTS

### 4.1 Experimental Settings

**Datasets.** The Ms-Celeb-1M [16], Labeled Face in the Wild (LFW) [21], and MegaFace challenge2 (MF2) [31] datasets are commonly used to assess the performance of FR models. We randomly select 400 individuals from the one-million celebrity list to form a Privacy-Celebrities dataset, where the identities are completely distinct from the ones in the Ms-Celeb-1M and LFW datasets. Each individual holds at least 20 face images. Among them, 10 images are for

training and others are for black-box testing. Additionally, the LFW dataset is utilized as our gallery set, which includes a total of 13,233 face images. Similarly, we randomly select 500 people from the MF2 dataset to build a Privacy-Commons dataset as the probe set, with a minimum of 10 images per identity. Another 10000 images selected from the MF2 dataset are combined into a corresponding gallery set. It should be noted that all input images are resized to $112 \times 112$.

**Target models.** In this paper, we employ the softmax loss, large margin cosine loss (LMCL) [38], and additive angular margin loss (ArcFace) [9] to fine-tune the pre-trained model (ResNet-50). The obtained three models are used to evaluate the performance of our scheme in the white-box settings and we denote them as S-ResNet, L-ResNet, and A-ResNet, respectively. Regarding the black-box setting, we choose another five models with diverse backbones and training losses, *i.e.*, Inception-ResNet [35], MobileNet [19], CosFace [38], SENet [20], and ArcFace [9]. These models are trained on the CAISA-WebFace dataset, which further simulates real-world scenarios.

**Compared Method.** To the best of our knowledge, there is no existing method to protect visual face privacy with a person-specific veil. Therefore, to demonstrate the superiority of our method, we additionally explore a feasible method for generating class-wise UAP, which is based on the combination concept of standard UAP and adversarial attack Type I. Specifically, we revise Eq. (1) and Eq (2) as

$$\underset{\delta}{\arg\max} \sum_{x_i^p \in X^p} d(x_i^p, x_i^{p\prime}),$$
$$s.t. \ \forall x_i^p \in X^p, \hat{F}(x_i^p) = \hat{F}(x_i^{p\prime}). \tag{15}$$

Equation (15) can be understood as aggregating vector-to-vector differences to craft the perturbations and we denote this method as V-UAP. Note that the process of generating the V-UAP excludes the use of block variance loss.

**Evaluation Metrics.** To assess the effectiveness of our scheme in concealing visual identity information, we utilize structural similarity index measure (SSIM) and learned perceptual image patch similarity (LPIPS) to measure the difference between the original images and the protected images. Both metrics aim to quantify how similar two images are perceived by human observers. A lower SSIM value indicates greater visual variations between two images, while a higher LPIPS value signifies a bigger visual perceptual difference. It is worth noting that the values of LPIPS are obtained by AlexNet [25].

Meanwhile, to evaluate the effectiveness of our scheme in retaining the functionality of FR, we perform the 1:N face identification experiments. In detail, we compare the outputs of the FR model on the original images and protected images and calculate the ratio that is referred to as the matching success rate (MSR) based on the matching outputs. The higher MSR indicates that our scheme has a slighter influence on the model's recognition accuracy.

## 4.2 Effectiveness of Visual Privacy Protection

*4.2.1 Effectiveness of Visual Anonymization.* In our scheme, each user can apply a person-specific veil to all his/her face images. Some protected images of different identities derived from Privacy-Celebrities are shown in Fig. 3. Obviously, the protected images

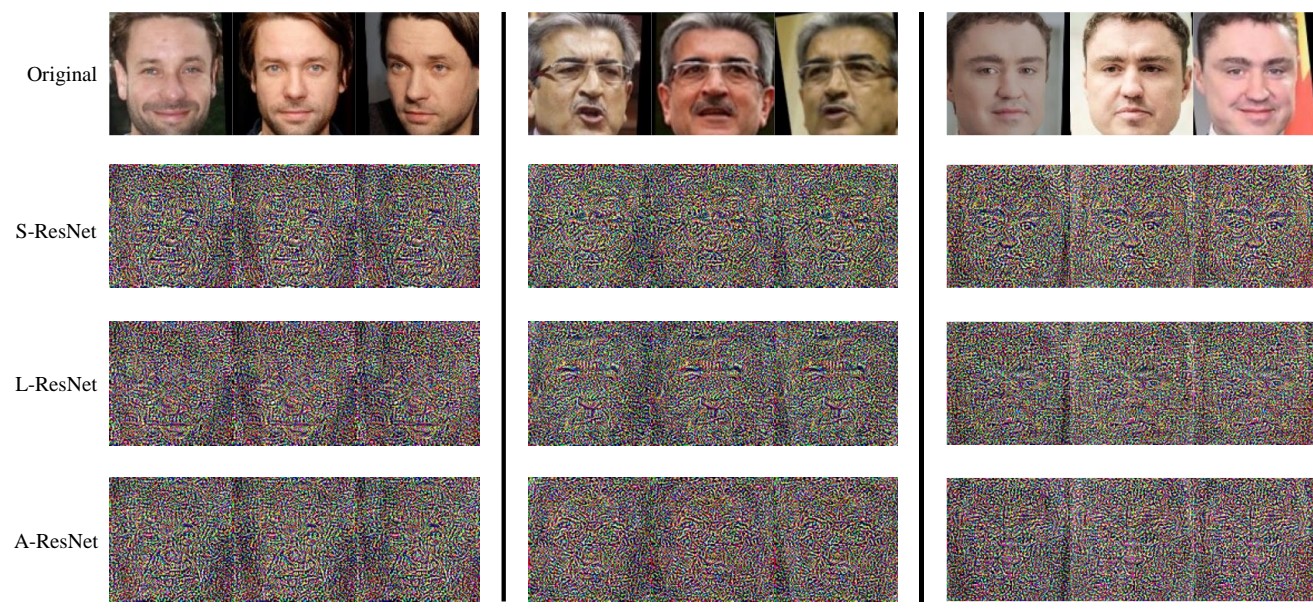

**Figure 3: Some protected images in the Privacy-Celebrities generated by S-ResNet, L-ResNet, and A-ResNet, respectively (under $\alpha = 0.2$). Each image within every subplot shares the same identity.**

**Table 1: The quantitative assessment of image visual quality using SSIM and LPIPS on the Privacy-Celebrities dataset.**

| Method | Source Model | SSIM(↓) | LPIPS (↑) |
|--------|--------------|---------|-----------|
| V-UAP | S-ResNet | 0.040 | 0.723 |
|  | L-ResNet | 0.043 | 0.710 |
|  | A-ResNet | 0.034 | 0.785 |
| Ours | S-ResNet | 0.011 | 1.250 |
|  | L-ResNet | 0.012 | 1.287 |
|  | A-ResNet | 0.011 | 1.247 |

generated by different models exhibit noticeable visual distinctions from the benign images and are confusing to human observers. Thus, it is challenging for human observers to extract meaningful visual identity information about the original images from the protected ones.

Moreover, for a better quantitative evaluation of the concealment of visual identity information, we calculate the SSIM and LPIPS values between the original and protected images. As shown in Table 1, the SSIM values between the protected and the original face images are comparatively low, whereas the LPIPS values are high. These quantitative results also indicate significant visual differences between the original and protected images.

**Compared with V-UAP.** In Table 1, it can be found that the SSIM values between the original images and the protected images generated with V-UAP are higher than ours (w/o) whereas the LPIPS values are lower. We infer that the identity feature subspace assigns weight coefficients to identity feature vectors and thus the crafted veils can more effectively alter the structure of the original images.

In summary, it can be concluded that the person-specific veils produced by our scheme are effective in visual anonymization.

### 4.2.2 Effectiveness of Identity Preservation.
We conduct 1:N face identification experiments to assess the degree to which protected images retain the functionality of FR. Specifically, we craft the protected images on the probe set (*e.g.*, Privacy-Celebrities) and then incorporate these images into the gallery set (LFW). Ultimately, we test the MSR of the protected images on several models. As shown in Fig. 4, when the testing model aligns with the surrogate model, the MSR reaches nearly 100%. It suggests that our scheme has a negligible impact on the training model's recognition accuracy.

In addition, we also consider a real-world situation where an attacker expects to identify the obtained protected images with a trained model. However, from the illustration of Fig. 4, the MSR decreases significantly even if the two models share the same backbone, especially when the surrogate models are S-ResNet and A-ResNet. It implies that even if attackers are perplexed by the protected images, they are tough to use their own trained model to obtain the true identity of the protected images. Likewise, we generate protected images using 10 images different from the training images and test their MSR across various models. Similar results are presented in the Appendix. In a nutshell, these quantitative results demonstrate the effectiveness of our scheme in preserving the functionality of FR.

**Table 2: The SSIM values and LPIPS values of recovered images on the Privacy-Celebrities dataset.**

| Method | Source Model | SSIM(↑) | LPIPS (↓) |
|--------|--------------|---------|-----------|
| ours | S-ResNet | 0.868 | 0.046 |
|  | L-ResNet | 0.869 | 0.047 |
|  | A-ResNet | 0.869 | 0.046 |

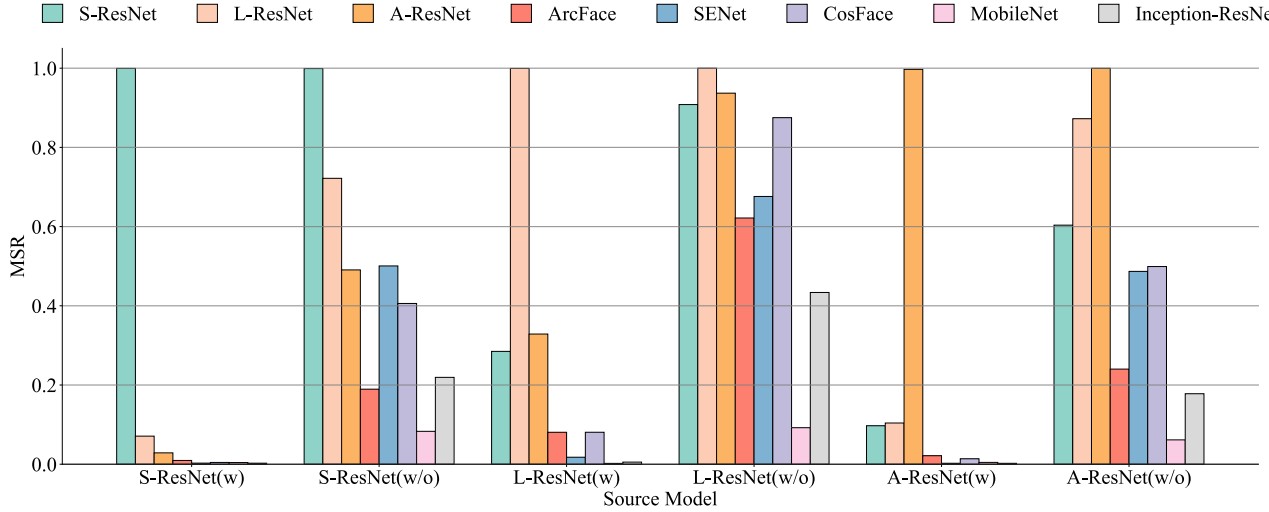

**Figure 4: The MSR of the protected images tested across different models on the Privacy-Celebrities dataset. 'w' and 'w/o' denote that the protected face images are generated with and without block variance loss, respectively.**

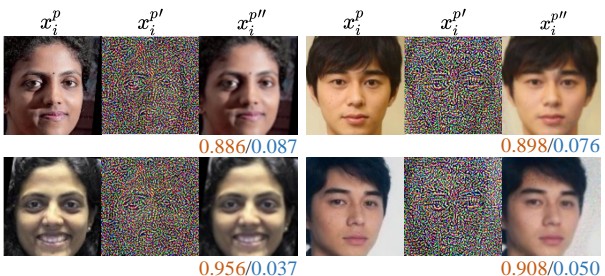

0.886/0.087     0.898/0.076

0.956/0.037     0.908/0.050

**Figure 5: Display of some original images ($x_i^p$), protected images ($x_i^{p'}$), and recovered images ($x_i^{p''}$). These protected images are generated by A-ResNet. The brown and blue numbers below ($x_i^{p''}$) are the SSIM and LPIPS values between ($x_i^p$) and ($x_i^{p''}$), respectively.**

*4.2.3 Evaluation of Supporting Recoverability.* In certain scenarios, authorized users anticipate obtaining the original images for various purposes. However, despite our scheme being theoretically fully reversible, practical usage constraints such as alpha values may hamper the original and recovered images from being exactly identical. Therefore, we endeavor to evaluate the performance of our scheme in supporting recoverability. As illustrated in Fig. 5, the recovered images closely resemble the originals visually. Furthermore, we calculate the SSIM and LPIPS values between them and the values are below the images. It can be observed that the SSIM values between $x_i^p$ and $x_i^{p''}$ are relatively high, while the LPIPS values are close to 0. Moreover, Table 2 provides comprehensive results of the recoverability performance. These experimental results indicate that our scheme is well-supportive of recoverability.

*4.2.4 Analysis of Efficient Protection.* In our experiment, generating a veil takes nearly 3 minutes on a GTX 3090 GPU with 24GB

RAM. Compared with [28, 33], our scheme crafts a person-specific veil with a limited number of face images. Once generated, such a veil can be conveniently overlaid on all images of the same user. It means that the generation time of veils is solely determined by the number of users but unaffected by the number of images. which is highly scalable and efficient. Especially, when dealing with a large set of face images, the superiority of our scheme in satisfying the requirements of real-time services will be presented. That is, our scheme takes an important step toward practical application in real-world scenarios.

### 4.3 Impact of the Parameter Tuning

Here, we mainly focus on assessing the impact of the parameter $\alpha$ on visual face privacy protection. We evaluate the performance across four different values of $\alpha$. As depicted in Fig. 6, the curves under these four parameter settings remain relatively stable, indicating that The variation of the alpha value within a certain range has a negligible impact on the MSR. However, it is noticeable from Fig. 8 that higher values of $\alpha$ result in more pronounced visual contours in the protected images. Based on our analysis, protected images generated with higher values are not suitable for visual face privacy protection. Taking into account both the visual effects of protected images and MSR, we selected $\alpha = 0.2$ to obtain satisfactory performance.

### 4.4 Robustness Analysis

Here, we employ Gaussian noise, salt and pepper noise as well as median filtering to the protected images to evaluate their robustness. The results tested across three models are presented in Fig. 7. It can be observed that when $\sigma$ exceeds 120, there is a noticeable decrease in the MSR of the protected images. It indicates that our scheme is robust to a certain level of noise. Additional details on the robustness analysis of our approach can be found in the appendix.

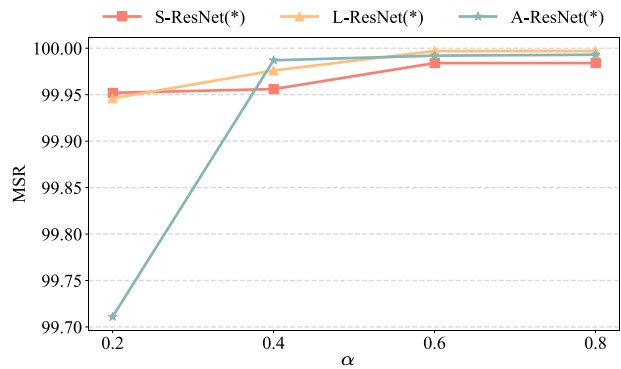

Figure 6: The impact of $\alpha$ on MSR of the protected images assessed on three surrogate models. '*' implies that the testing model is identical to the surrogate model.

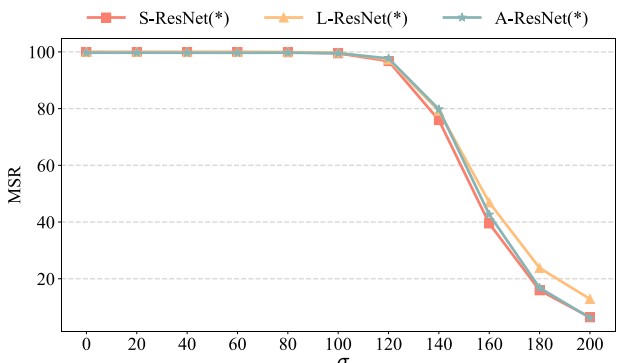

Figure 7: Robustness testing of protected images against Gaussian noise. $\sigma$ denotes the standard deviation of the added noise.

Table 3: The quantitative results of the ablation study. 'w' and 'w/o' denote that the protected face images are generated with and without block variance loss, respectively.

| Method | Source Model | SSIM(↓) | LPIPS (↑) |
|---|---|---|---|
| Ours (w) | S-ResNet | 0.011 | 1.250 |
| | L-ResNet | 0.012 | 1.287 |
| | A-ResNet | 0.011 | 1.247 |
| Ours (w/o) | S-ResNet | 0.032(+0.021) | 0.776(-0.474) |
| | L-ResNet | 0.037(+0.025) | 0.734(-0.553) |
| | A-ResNet | 0.030(+0.019) | 0.852(-0.395) |

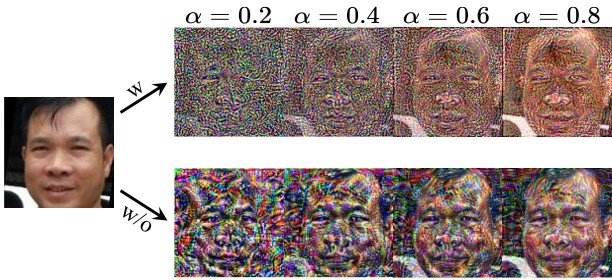

Figure 8: The influence of different $\alpha$ on the visual presentation of the protected images. 'w' and 'w/o' denote that the protected face images are generated with and without block variance loss, respectively.

## 4.5 Ablation Study

We craft two categories of veils: one with block variance loss and the other without, to generate corresponding protected images, respectively. From the illustration in Fig. 8, it is evident that the protected images generated without block variance loss retain more visual contour information compared to those that do. According to the visual bias in the human visual perception system, observers tend to focus more on the shape or size dimensions of an object rather than other perceptual dimensions [27]. It suggests that the above contour information can also be linked to identity, which may still lead to visual privacy leakage. Meanwhile, as shown in Fig. 4, these protected images have higher MSR on other models, which implies that there is an increased likelihood for attackers to obtain real visual identities. Furthermore, Table 3 displays that the protected images generated without block variance loss have higher SSIM and lower LPIPS values. These results quantitatively demonstrate that the protected images generated without block variance loss are visually closer to the original images. Therefore, it can be concluded that the utilization of block variance loss is crucial during the generation of person-specific veils.

## 4.6 Discussion

**Person-specific vs Universal.** Compared to person-specific veils, generating UAPs will be more efficient. However, since each user possesses unique visual and identifiable information, the UAP-based

methods fail to provide effective visual face privacy protection in the conflict between universality and uniqueness. In other words, person-specific veils provide a better trade-off between them to the users. Furthermore, a person-specific veil is only applicable to an identity while a UAP is applicable to all. Accordingly, when being attacked, the UAP can be removed from all the protected images of different identities whereas the adversary can only remove the veil from the protected images of a single identity. It means that the person-specific veil offers better security compared with UAPs.

## 5 CONCLUSION

In this paper, we develop an efficient visual face privacy protection scheme by utilizing person-specific veils. A user can conveniently apply his/her veil to all images of him/her to generate the protected images without crafting new perturbations for each one. These protected images are significantly different from the originals in visual but can retain the functionality of FR. Moreover, Our method promotes the alignment between the recognition outputs of protected and original images by constructing the feature subspace and enhances the concealment of visual identity information with block variance loss. Extensive experiments demonstrate that our scheme achieves satisfactory performance on visual anonymization and identity preservation.

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
