# OpenReview forum: "Once-for-all: Efficient Visual Face Privacy Protection via Person-specific Veils"
_acmmm.org/ACMMM/2024/Conference — MM2024 Oral_

### Official Review · Reviewer_BKSv · 2024-05-17

**Rating:** 6
**Confidence:** 4

**Summary:**

This paper focuses on the issue of visual facial privacy leaks in cloud platforms and aims to efficiently deal with it. The authors present a novel visual face privacy scheme by using the person-specific veils, which breaks the limitations in existing schemes. This sort of veil only needs to be generated once and can be applied to all images of a user. To generate it, they propose a method that consists of visual transformation and identity preservation.

**Strengths:**

1.Different from the typical adversarial perturbation, the utilization of significant perturbation to protect visual face privacy is innovation and attractive.
2.The motivation of this paper is sound, and the use of class-wise universal adversarial perturbation to protect personal images is understandable.
3.The experiments are clear, the results are effective.
4.The paper is well-organized with a coherent structure, making it easy to grasp the main ideas and viewpoints.

**Limitations:**

1.In lines 144-146, the authors say “Compared with previous schemes, our scheme takes an important step toward a practical system with real-time requirements”. What does 'real-time' here specifically refer to? Please explain it based on the specific application scenario.
2.In Figure 4, the MSR corresponding to models other than the agent model is low. Does this mean that the proposed method has poor transferability? The authors should explain this in the context of the availability of protected images.
3.Please carefully check the format of mathematical symbols in the text. For example, matrices and vectors should be bold, and sets should be hollow.

**Suitability:**

3

---

### Official Review · Reviewer_KhSb · 2024-05-23

**Rating:** 6
**Confidence:** 4

**Summary:**

This paper proposes an efficient visual face privacy protection scheme by using the person-specific veils. To generate this type of veil, the authors propose a novel method consisting of two parallel optimization objectives, namely visual transformation and identity preservation. Interestingly, the generation is only once but the veil can be applied to all images of the same user. Empirical results show the effectiveness of the proposed scheme on two datasets.

**Strengths:**

1. The research motivation for efficient visual face privacy protection is practically significant.
2. The using of person-specific veil is innovative and intriguing.
3. The framework of proposed method is straightforward and clean. The reason for using each loss function is presented well.
4. The paper is well -written, the charts are clear and easy to understand. Additionally, extensive experiments and analysis are provided.

**Limitations:**

1. I'm curious whether there is only one such veil for a single user. If not, what is the difference in the protection effect it offers compared to diversified perturbations? If there is, does that mean the scheme has lower security? Please provide further explanation of the scheme's security.
2. For constructing the identity feature subspace, the number and variety of face images selected seem to have a significant impact. Intuitively, fewer images and single postures result in a less effective space, whereas more images and diverse postures yield better results. How can one choose the number of face images and the variety of facial postures in conjunction with real-world scenarios?
3. The recognition performance before and after protection needs to be presented to highlight the impact on the face recognition models.

**Suitability:**

2

---

### Official Review · Reviewer_hjsQ · 2024-05-23

**Rating:** 5
**Confidence:** 3

**Summary:**

The paper presents an efficient visual face privacy protection scheme by utilizing person-specific veils, which can be conveniently applied to all images of the same user without regeneration.

**Strengths:**

The notation of using person-specific veils in facial privacy protection is quite new and interesting. The overall novelty of the paper is high. The proposed method is technically sound.

**Limitations:**

The clarity of the paper presentation and some details can be improved. For example, more details about how V-UAP is implemented can be given.

It is better to make a comparison with the work “Hiding visual information via obfuscating adversarial perturbations”published on ICCV 2023.

Qualitative comparison results against V-UAP with image examples could be provided.

Some minor mistakes exist, e.g., Formula (8) seems to have a redundant left bracket. I suggest the authors carefully check every single formula and the consistencies.

**Suitability:**

3

---

### Official Review · Reviewer_3R6r · 2024-05-25

**Rating:** 4
**Confidence:** 4

**Summary:**

The paper proposes an effective visual face privacy protection scheme, which can generate a specific veil for each user to protect privacy. Here a user can apply a specific veil to protect all his/her images. It is an interesting work. The experimental results prove the effectiveness of the proposed scheme.

**Strengths:**

1. The paper proposes an effective visual face privacy protection scheme, generating a dedicated veil for each user, and reducing computational consumption.
2. The idea of utilizing person-specific veils for visual privacy protection is interesting.
3. The experimental results show that the proposed method can protect visual features and achieve identity preservation.
4. The organization of the paper is well and easy to read.

**Limitations:**

1. The comparison experiments are not sufficient. The authors show the experimental results of different target models, but lack more comparison methods, like AVIH [1].
2. The block variance constraint is also leveraged in AVIH. Please clearly illustrate the differences and novelty in this paper.
3. The ablation study only evaluates the block variance loss, which is not sufficient. Authors should verify the mentioned novel components here.
4. How about evaluating the proposed model with different recognition models in the testing? I think it is essential to evaluate its robustness ability.
5. Since the same veil is used for all the images of that user, is it easy for an attacker to crack that veil?
6. In section 4.3, the authors explore the impact of parameters on privacy protection, but the identification assessment is missing.
References:
[1] Hiding Visual Information via Obfuscating Adversarial Perturbations, International Conference on Computer Vision (ICCV), 2023.

**Suitability:**

3

---

### Meta-Review · Area_Chair_twRP · 2024-06-29

**Recommendation:** Accept (Oral)
**Confidence:** 5

**Metareview:**

The submission received two accepts, one weak accept and one borderline accept. Reviewers in general appreciated the novelty and experiments of the work. Concerns were in general addressed well by the rebuttal. With all the information into consideration, we agree with the reviwers and would like to recomemnd the work to ACM MM.